# Anti-*Toxoplasma gondii* IgG seroprevalence in the general population in Iran: A systematic review and meta-analysis, 2000–2023

**Faezeh Hamidi[1], Ali Rostami[2], Seyed Abdollah Hosseini[3,4], Rafael Calero-Bernal📷[5], Jafar Hajavi[6], Reza Ahmadi[6], Hossein Pazoki📷[6]***

1 Faculty of Medical Sciences, Department of Laboratory Sciences and Microbiology, Tabriz Medical Sciences, Islamic Azad University, Tabriz, Iran, 2 Health Research Institute, Infectious Diseases and Tropical Medicine Research Centre, Babol University of Medical Sciences, Babol, Iran, 3 Communicable Disease Institute, Toxoplasmosis Research Center, Mazandaran University of Medical Sciences, Sari, Mazandaran, Iran, 4 Department of Parasitology and Mycology, School of Medicine, Mazandaran University of Medical Science, Sari, Iran, 5 Veterinary Faculty, Animal Health Department, SALUVET, Complutense University of Madrid, Madrid, Spain, 6 Faculty of Medicine, Department of Medical Microbiology, Infectious Diseases Research Center, Gonabad University of Medical Science, Gonabad, Iran

* hosseinpazoki11@gmail.com

**Data Availability Statement:** The paper and its Supporting Information files contain all the raw data required to replicate the results of this study.

## Abstract

Toxoplasmosis ranks among the most prevalent parasitic diseases globally. It seems that chronic toxoplasmosis is associated with several neuropsychiatric and other harmful effects in infected people, therefore, there is a need to investigate the prevalence of toxoplasmosis across various world regions. In this study, we conducted a meticulous meta-analysis to estimate the seroprevalence of anti-*Toxoplasma gondii* IgG antibodies within the general population in Iran (GPI). International and national scientific databases for studies published between January 1, 2000, and September 30, 2023, were searched. Observational studies reporting anti-*T. gondii* IgG seroprevalence in the GPI was selected/included. The data were synthesized using a random-effects model to calculate with a 95% confidence interval (95% CI) the national and regional anti-*T. gondii* IgG seroprevalence rates in Iran. Additionally, subgroup analyses were conducted to investigate the frequency of exposition to *T. gondii* in different socio-demographic, climatic, and geographical scenarios. From 18661 identified studies, 327 were included in the present meta-analysis, encompassing 122,882 individuals across the 31 Iranian provinces. The pooled nationwide anti-*T. gondii* IgG seroprevalence among the GPI was determined to be 32.9% (95% CI: 30.9–35.1%). The highest anti-*T. gondii* IgG seroprevalence was observed in Mazandaran province (North of Iran) (61%), whereas the lowest was in Semnan province (12.5%).Anti-*T. gondii* IgG seroprevalence demonstrated a higher occurrence in provinces characterized by moderate temperatures of 16–21˚C, high relative humidity, and annual precipitation. Additionally, a higher anti-*T. gondii* IgG seroprevalence was identified among individuals with a habit of consumption of undercooked meat, raw fruits or vegetables, and untreated water. Moreover, those reporting direct contact with cats, possessing a lower level of education, residing in rural areas, being engaged in farming occupations, or playing the role of housewives exhibited higher anti-*T. gondii* IgG seroprevalence figures.The anti-*T. gondii* IgG seroprevalence within GPI

**Funding:** The author(s) received no specific funding for this work.

**Competing interests:** The authors have declared that no competing interests exist.

closely aligns with the estimated worldwide average exposition rates. This underscores the imperative for public health policymakers to prioritize educational efforts regarding toxoplasmosis transmission pathways and its link to harmful effects.

## Introduction

*Toxoplasma gondii* is an intracellular opportunistic protist estimated to infect approximately one-third of the human global population. However, regions in Africa, South America, and Southeast Asia exhibit notably high seroprevalence rates [1]. Immunocompromised individuals such as organ transplant recipients, HIV-positive individuals, and cancer patients may face potentially life-threatening infections due to parasite reactivation. In HIV/AIDS patients, repeated bouts of encephalitis may occur, leading to long-term neurological harm and even death [2–4].

Additionally, congenital toxoplasmosis after transplacental transmission of the parasite and fetus invasion represents another remarkable clinical manifestation, contributing to abortion, mental retardation, seizures, and hearing or visual impairments in affected neonates [5]. Moreover, this parasite of neurotropic nature has been linked to psychiatric and behavioral disorders during chronic infections [6]. It has been observed correlation between chronic toxoplasmosis infection and psychiatric disorders[7–9] underscoring the importance of reliable diagnosis of chronic toxoplasmosis.

The exposition of *T. gondii* in the Iranian population has been mostly approached by the detection of anti-*Toxoplasma* IgG by serological methods like enzyme-linked immunosorbent assay (ELISA), and indirect immunofluorescence antibody test (IFA) [10,11]. While several meta-analysis studies focused on *T. gondii* occurrence in immunocompromised individuals such [12,13], limited evidence exists regarding the overall anti-*T. gondii* IgG seroprevalence within the GPI. Previously, only one meta-analysis study on a limited set of articles (35 records) had been published [14]; and a more exhaustive data compilation is needed to clarify the epidemiological scenario better and justify intervention strategies against toxoplasmosis.

Herein we present a timely systematic review and meta-analysis to assess the anti-*T. gondii* IgG seroprevalence in the GPI, along with evaluation of the potential population risk factors and geo-climatic conditions that may influence *T. gondii*-exposition levels in the GPI.

## Methods

### Search strategy and study selection

We have conducted and reported this review study following the PRISMA (Preferred Reporting Items for Systematic Reviews and Meta-Analyses) guidelines [15]. It should also be mentioned that the ethics committee of Gonabad University of Medical Sciences (IR.GMU. REC.1402.130) approved the protocol of this study. In September 2023, two independent investigators (JH and FH) systematically searched international and national scientific databases, including PubMed/MEDLINE, Scopus, Science Direct, Web of Science, MagIran, and the Scientific Information Database (SID) for articles published between 2000 and 2023. Google Scholar and reference lists of relevant review papers were also searched for "gray literature". The search terms used encompassed ((*Toxoplasma*) OR (*Toxoplasma gondii*) OR (*T. gondii*) OR (Toxoplasmosis)) AND ((Iran) OR (Islamic Republic of Iran)) AND ((seroprevalence) OR (seropositivity) OR (Prevalence)). All observational studies cover abstracts, full texts, and

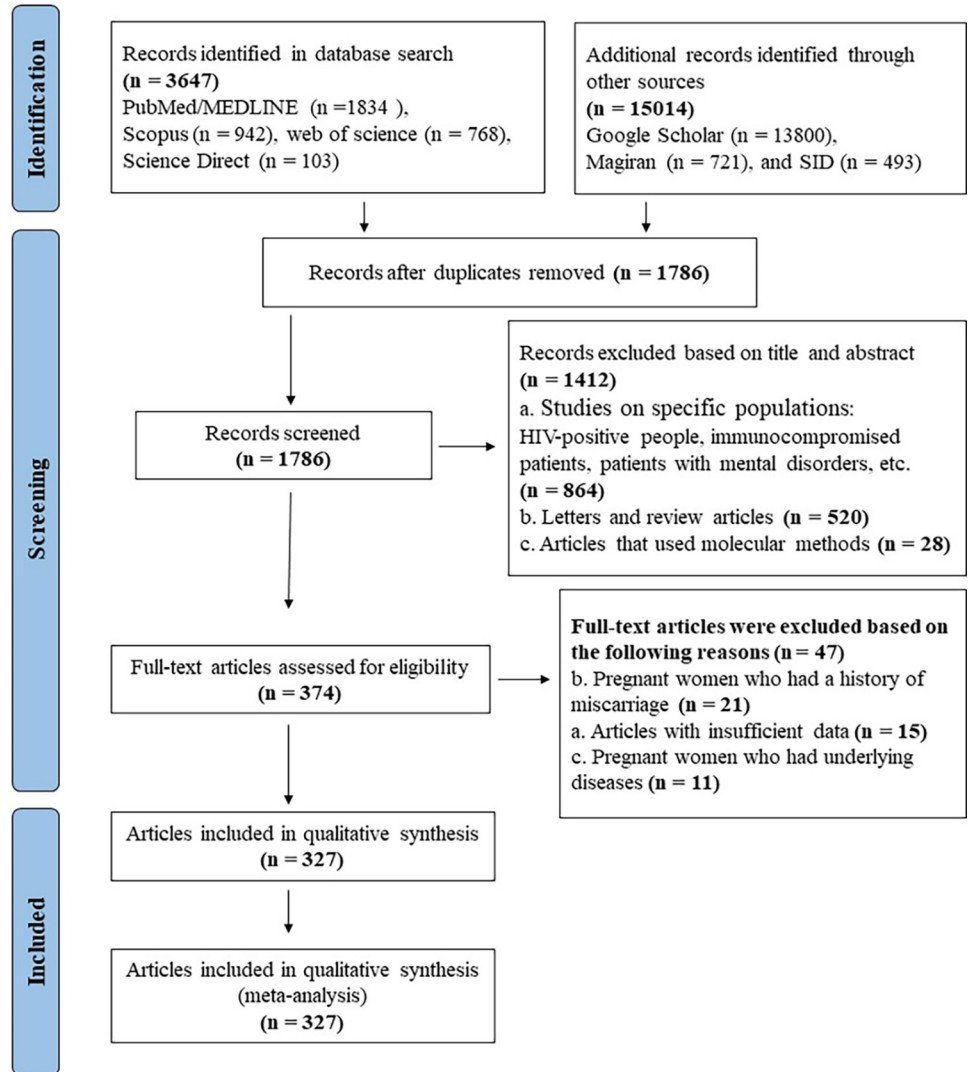

**Fig 1. Flowchart of the study design process.**

proceedings from national conferences reporting anti-*T. gondii* IgG seroprevalence in the GPI was included (Fig 1). Our search was limited to articles in Persian and English languages. Additionally, studies involving pregnant women without a history of abortion were considered. However, we excluded duplicated studies, those utilizing non-random sampling methods, letters, review articles, studies lacking sufficient data, those employing molecular methods, and pregnant women with underlying diseases. Special population segments/types, such as HIV-positive individuals, immunocompromised patients, and those with mental disorders, were also excluded. Nevertheless, if these studies adopted a case-control design, the control group, comprising healthy individuals, was included for analysis.

After eliminating duplicate studies, two researchers (SAH and AR) screened titles and abstracts. Subsequently, the full texts of relevant studies underwent screening, and any disparities in study selection between the two investigators were resolved through consensus by a third investigator.

## Data extraction

Information extracted from the eligible studies encompassed several data variables: the last name of the first author, year of publication, title, population type, study design, quality assessment, location, diagnostic method, sample size, number of IgG-positive cases, number of male and female participants, residency details, age distribution, education level, occupation, cat contact, consumption of raw meat, raw vegetables and fruits, washing practices for vegetables and fruits, and drinking water source. Additionally, we retrieved details regarding longitude, latitude, average temperature, average relative humidity, and precipitation rate for each location/municipality under study. These data were sourced from websites such as timeanddate. com (accessible at: https://www.timeanddate.com/weather/iran/tehran/climate) and climate-data.org (accessible at: https://en.climate-data.org/).

## Quality assessment

The Joanna Briggs Institute's (JBI) critical appraisal checklist for studies reporting prevalence data was utilized to evaluate the quality of the studies [16]. Two authors (HP and FH) performed the assessment, and disagreements between them were resolved by discussion. The evaluation tool included several questions about the quality of the study, and based on these questions, the studies were scored. Finally, total quality scores of $\geq 3$, 4 to 6, and $7 \leq$ were considered low, medium, and high quality, respectively.

## Meta-analyses

All statistical analyses were conducted using the StatsDirect version 3.0 statistical analysis software (StatsDirect.exe; https://www.statsdirect.com/). The heterogeneity among studies was assessed using Cochran's Q test and the $I^2$ statistic, with a significance level set at 50% to define a statistically significant degree of heterogeneity. The random-effect model (REM) was employed to calculate the pooled anti-*T. gondii* IgG seroprevalence and corresponding 95% confidence intervals (CI) [17]. We stratified anti-*T. gondii* IgG seroprevalence is based on the 31 provinces in Iran. We performed several subgroup analyses to compare the anti-*T. gondii* IgG seroprevalence according to the following parameters: type of study, risk of population (healthy people, pregnant women, blood donors), location, diagnostic method, sample size, and positive cases based on gender, place of residence, age distribution, education level, occupation, contact with cats, consumption of raw (undercooked) meat, raw vegetables and fruits, washing vegetables and fruits prior intake, source of drinking water. We also analyzed the geographical distribution of studies (north, center, west, and east of Iran), longitude, latitude, mean annual temperature, relative humidity, and precipitation [18]. Furthermore, publication bias in the selected studies was evaluated using a funnel plot, Egger's test, and Harbord bias [19].

## Results

Our initial database search generated 18,661 publications, with subsequent screening based on titles, abstracts, and full texts, which excluded 18,334 articles. Ultimately, 327 articles met the inclusion criteria and were selected for the meta-analysis (Fig 1). These studies tested 122,882 individuals of GPI across all 31 provinces. A very high degree of heterogeneity among the included studies (Q = 20587.3, $I^2$ = 98.4%, P<0.001) is observed, and therefore/in consequence, a REM was employed to estimate the more conservative anti-*T. gondii* IgG seroprevalence record among the GPI. The pooled nationwide anti-*T. gondii* IgG seroprevalence among the GPI was determined to be 32.9% (95% CI: 30.9–35.1%). Furthermore, neither the funnel

**Table 1. Subgroup analysis of the pooled seroprevalence of anti-*Toxoplasma gondii* IgGs in the general population of Iran stratified by province (2000 to 2023).**

| Province | No. datasets | No. samples | No. seropositive | Pooled prevalence % (95% CI) | Heterogeneity (Q) | $I^2$ (%) |
|---|---|---|---|---|---|---|
| Alborz | 4 | 1611 | 393 | 24.0 (22.0–26.0) | 28.4 | 89.4 |
| Ardabil | 6 | 2396 | 616 | 25.0 (23.0–26.0) | 214.6 | 97.7 |
| Bushehr | 5 | 1918 | 453 | 20.0 (14.0–30.0) | 50.7 | 94.1 |
| Chaharmahal and Bakhtiari | 14 | 3946 | 1379 | 35.0 (28.0–39.0) | 175.3 | 92.6 |
| East Azerbaijan | 23 | 10899 | 3820 | 46.0 (39.0–53.0) | 917.2 | 97.6 |
| Fars | 16 | 8374 | 968 | 14.0 (11.0–17.0) | 242.9 | 93.8 |
| Golestan | 7 | 2696 | 1173 | 42.0 (35.0–50.0) | 107.5 | 94.4 |
| Guilan | 3 | 1100 | 454 | 54.0 (29.0–77.0) | 90.8 | 97.8 |
| Hamedan | 7 | 5655 | 1747 | 33.0 (26.0–41.0) | 173.9 | 96.6 |
| Hormozgan | 4 | 1629 | 568 | 34.0 (27.0–40.0) | 23.3 | 87.1 |
| Ilam | 6 | 2078 | 616 | 29.0 (18.0–40.0) | 138.4 | 96.4 |
| Isfahan | 17 | 6815 | 2379 | 35.0 (30.0–41.0) | 344 | 95.3 |
| Kerman | 9 | 2998 | 802 | 22.0 (13.0–32.0) | 317.7 | 97.5 |
| Kermanshah | 6 | 6406 | 1503 | 24.0 (15.0–34.0) | 368.5 | 98.6 |
| Khuzestan | 39 | 6955 | 1867 | 26.0 (23.0–30.0) | 444.2 | 91.4 |
| Kohgiluyeh and Boyer-Ahmad | 4 | 1764 | 266 | 14.0 (10.0–19.0) | 19.4 | 84.5 |
| Kurdisatn | 7 | 1436 | 360 | 23.0 (18.0–29.0) | 32.2 | 81.4 |
| Lorestan | 10 | 3742 | 1837 | 44.0 (29.0–60.0) | 806.9 | 98.9 |
| Markazi | 10 | 1892 | 635 | 37.0 (31.0–43.0) | 59.4 | 84.9 |
| Mazandaran | 25 | 9753 | 5411 | 61.0 (52.0–71.0) | 2271.8 | 98.9 |
| North Khorasan | 4 | 2228 | 866 | 26.0 (7.0–52.0) | 403.4 | 99.3 |
| Qazvin | 3 | 850 | 411 | 48.0 (45.0–52.0) | 64.5 | 96.9 |
| Qom | 9 | 2332 | 1076 | 47.0 (45.0–52.0) | 409.9 | 98 |
| Razavi Khorasan | 16 | 4126 | 1144 | 27.0 (22.0–32.0) | 199.5 | 92.5 |
| Semnan | 1 | 400 | 50 | 12.5 | - | - |
| Sistan and Baluchestan | 15 | 3919 | 900 | 25.0 (21.0–29.0) | 110.5 | 87.3 |
| South Khorasan | 4 | 1543 | 283 | 19.0 (10.0–30.0) | 75.2 | 96 |
| Tehran | 33 | 16141 | 6841 | 33.7 (25.4–42.7) | 4008.7 | 99.2 |
| West Azerbaijan | 9 | 2599 | 801 | 33.0 (26.0–41.0) | 130.1 | 93.9 |
| Yazd | 6 | 1722 | 337 | 16.8 (9.8–25.2) | 87.4 | 94.3 |
| Zanjan | 1 | 500 | 186 | 37.2 | - | - |
| Not determined | 4 | 1199 | 564 | 30.0 (9.0–57.0) | 169 | 98.2 |
| Total (Iran) | 327 | 121622 | 40706 | 32.9 (30.9 - 35.1%) | 20587.3 | 98.4 |

plot nor the bias coefficient diagram indicated the presence of publication bias (Harbord bias = -0.07, Egger's test = 3.73, P = 0.939) (S1 Fig in S1 File).

## Subgroup analysis on environmental and diagnostic variables

Notably, northern Mazandaran (61%) and Gilan (54%) provinces exhibited the highest anti-*T. gondii* IgG seroprevalence rates. In contrast, Semnan (12.5%), Fars (14%), and Kohgiluyeh and Boyer-Ahmad (14%) provinces, located in the central areas, demonstrated the lowest anti-*T. gondii* IgG seroprevalence rates (Table 1, Fig 2). The anti-*T. gondii* IgG seroprevalence across different periods revealed varying rates, with the highest prevalence observed between 2011 and 2020 at 33.4% (31.0–35.9%). In contrast, the prevalence was slightly lower for years preceding 2010 at 33.0% (27.0–39.0%) and for years post-2021 at 31.0% (24.0–37.0%) (Table 2).

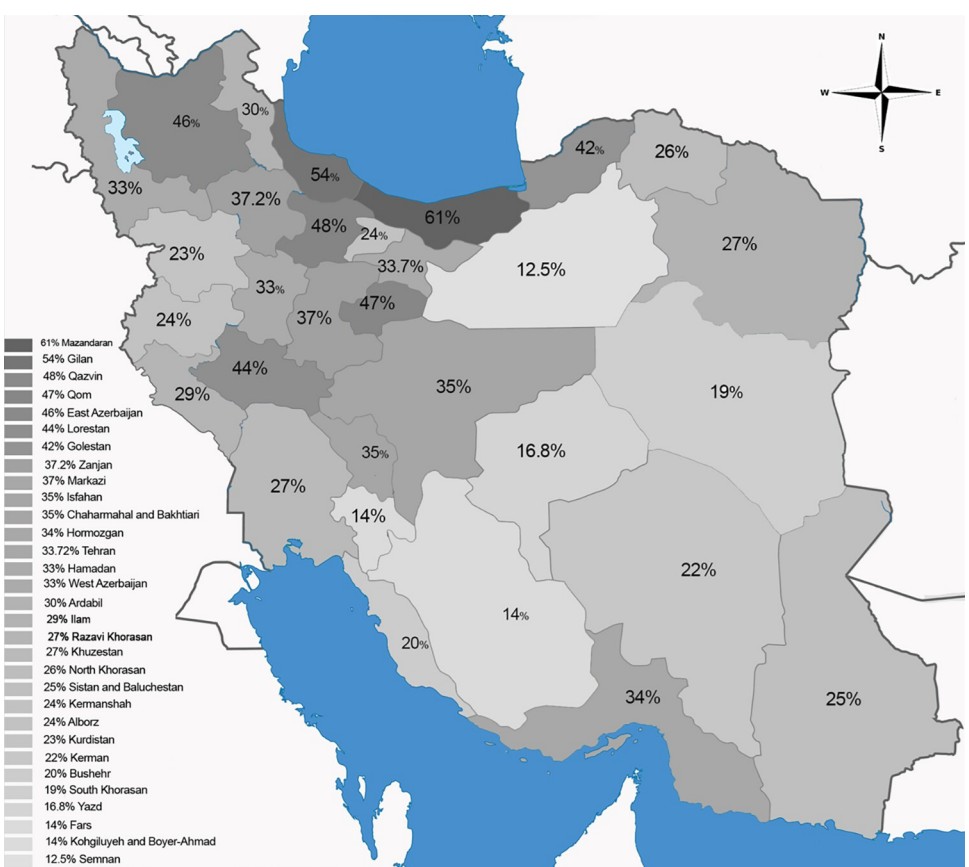

**Fig 2. Seroprevalence of anti-*Toxoplasma gondii* IgG in the general population residing in each of the 31 provinces of Iran.** Geographic information systems have been used for data representation (ArcGIS software version 10.2.2; ESRI, Redlands, California, USA).

Regarding diagnostic methods, the highest anti-*T. gondii* IgG seroprevalence was reported using the IFAT method, reaching 40.0% (95% CI: 33.0–48.0%). In comparison, the anti-*T. gondii* IgG seroprevalence for CLIA (Chemiluminescence immunoassay) and ELISA methods was 36% (250–48.0%) and 31.9% (29.9–34.0%), respectively (Table 2). Statistically significant differences were observed in all subgroups of environmental and diagnostic variables (P<0.001) (Table 2).

An analysis based on geo-climatic parameters aimed to uncover sources of heterogeneity affecting anti-*T. gondii* IgG seroprevalence in the GPI. Subgroup analyses based on geographical longitude revealed the highest anti-*T. gondii* IgG seroprevalence within the 35–38 N latitude at 39.4% (35.6–43.2%). Comparatively, 30 – 35N and 27 – 30N latitudes exhibited anti-*T. gondii* IgG seroprevalence rates of 30.8% (95% CI: 27.8–33.8%) and 22.0% (95% CI: 19.0–25.0%), respectively (Table 2). Similarly, considering geographical latitude, regions within 44 – 49E and 49 – 54E longitude displayed anti-*T. gondii* IgG seroprevalence rates of 33.0% (30.0–36.0%) and 36.2% (31.9–40.7%), respectively. Contrastingly, areas within 54 – 59E and >59E showed lower rates at 24.6% (19.4–30.3%) and 27.0% (24.0–31.0%), respectively (Table 2).

Analysis based on mean temperature revealed a higher anti-*T. gondii* IgG seroprevalence in regions with temperatures between 16–21˚C, with a prevalence of 35.8% (32.3–39.5%) compared to areas with temperatures of 10–15˚C and ≥22˚C (Table 2). Concerning humidity, regions with humidity levels ≥76% exhibited the highest anti-*T. gondii* IgG seroprevalence at 58.0% (46.0–70.0%). Regions with 36–55% and 56–75% humidity levels showed anti-*T. gondii*

**Table 2. Subgroup analysis of anti-*Toxoplasma gondii* IgGs seroprevalence within the general population of Iran according to environmental and diagnostic variables (2000 to 2023).**

| Subgroup variable | No. samples (%) | No. seropositive | Prevalence % (95% CI) | Heterogeneity (Q) | $I^2$ (%) | Interaction test ($X^2$ value) | P-value |
|---|---|---|---|---|---|---|---|
| **Year** | | | | | | 25.23 | <0.001 |
| ≤ 2010 | 26621 | 10349 | 33.0 (27.0–39.0) | 5059.1 | 99.0 | | |
| 2011–2020 | 85641 | 27093 | 33.4 (31.0–35.9) | 12931.5 | 98.2 | | |
| ≥2021 | 10620 | 3264 | 31.0 (24.0–37.0) | 2157.1 | 98.1 | | |
| **Longitude** | | | | | | 1919.07 | <0.001 |
| 27–30 | 17876 | 3710 | 22.0 (19.0–25.0) | 1439.4 | 96.4 | | |
| 30–35 | 38224 | 11794 | 30.8 (27.8–33.8) | 4390.9 | 97.5 | | |
| 35–38 | 49357 | 18693 | 39.4 (35.6–43.2) | 9151.6 | 98.6 | | |
| **Latitude** | | | | | | 536.59 | <0.001 |
| 44–49 | 41835 | 13056 | 33.0 (30.0–36.0) | 4406.2 | 97.3 | | |
| 49–54 | 47004 | 16594 | 36.2 (31.9–40.7) | 10870.1 | 98.9 | | |
| 54–59 | 9590 | 2850 | 24.6 (19.4–30.3) | 1236.7 | 97.6 | | |
| >59 | 7028 | 1697 | 27.0 (24.0–31.0) | 211.2 | 89.1 | | |
| **Mean temperature** | | | | | | 12447.75 | <0.001 |
| 10–15˚C | 29763 | 9497 | 35.0 (32.0–38.0) | 2072.1 | 96.2 | | |
| 16–21˚C | 56363 | 20899 | 35.8 (32.3–39.5) | 11798.2 | 98.8 | | |
| ≥22˚C | 18071 | 3811 | 23.0 (20.0–26.0) | 1447.5 | 95.5 | | |
| **Humidity** | | | | | | 2168.12 | <0.001 |
| ≤ 35% | 8832 | 1985 | 21.3 (17.4–25.5) | 565.2 | 95.0 | | |
| 36–55% | 79997 | 2613 | 32.0 (29.4–34.7) | 12632.8 | 98.4 | | |
| 56–75% | 10949 | 3409 | 32.0 (27.0–37.0) | 1024.4 | 97.2 | | |
| ≥76% | 5679 | 2690 | 58.0 (46.0–70.0) | 1859.7 | 98.9 | | |
| **Precipitation rate** | | | | | | 2655.61 | <0.001 |
| ≤10 | 50928 | 16117 | 30.2 (26.9–33.6) | 8219.6 | 98.5 | | |
| 11–20 | 15266 | 4646 | 30.0 (26.0–34.0) | 1790.8 | 96.6 | | |
| 21–30 | 29506 | 8418 | 30.0 (27.0–33.0) | 2421.0 | 97.1 | | |
| 31–40 | 3021 | 1587 | 46.0 (29.0–63.0) | 724.1 | 98.9 | | |
| ≥41 | 6736 | 3429 | 59.0 (47.0–70.0) | 2050.7 | 98.9 | | |
| *Diagnostic methods | | | | | | | <0.001 |
| IFA | 19519 | 9298 | 40.0 (33.0–48.0) | 4496.4 | 99.2 | 488.06 | |
| ELISA | 99618 | 30044 | 31.9 (29.9–34.0) | 13489.3 | 97.9 | | |
| CLIA | 2395 | 777 | 36.0 (25.0–48.0) | 178.5 | 96.6 | | |

\* IFA: Indirect immunofluorescence Antibody, ELISA: Enzyme-Linked Immunosorbent Assay, CLIA: Chemiluminescence immunoassay.

IgG seroprevalence of 32.0% (29.4–34.7%) and 32.0% (27.0–37.0%), respectively. In contrast, regions with humidity levels ≤35% displayed the lowest anti-*T. gondii* IgG seroprevalence at 21.3% (17.4–25.5%) (Table 2). Analysis based on precipitation showed a significantly higher anti-*T. gondii* IgG seroprevalence in areas with precipitation ≥41 millimeters 59.0% (47.0–70.0%). Conversely, regions with precipitation 31–40, ≤10, 21–30, and 11–20 millimeters exhibited descending prevalence rates of 46.0% (29.0–63.0%), 30.9% (26.9–33.6%), 30.0% (27.0–33.0%), and 30.0% (26.0–34.0%), respectively (Table 2).

## Subgroup analysis on demographic variables

Upon conducting a subgroup analysis based on age groups, the anti-*T. gondii* IgG seroprevalence was notably higher among individuals aged ≥ 50 years, reaching 44.4% (31.9–57.2%). Subsequently, those aged 21–50 years showed an anti-*T. gondii* IgG seroprevalence of 36% (31–4%), followed by 11-20-year-old at 25.6% (21.7–29.9%), while children under 10 exhibited the lowest anti-*T. gondii* IgG seroprevalence at 17.1% (11.9–23.1%) (Table 3). Analyzing educational levels, the highest anti-*T. gondii* IgG seroprevalence was observed among illiterate

**Table 3. Subgroup analysis of anti-*Toxoplasma gondii* IgGs seroprevalence in the general population of Iran according to demographic variables (2000 to 2023).**

| Subgroup variable | No. samples | No. seropositive | Prevalence % (95% CI) | Heterogeneity (Q) | $I^2$ (%) | Interaction test ($X^2$ value) | P-value |
|---|---|---|---|---|---|---|---|
| **Gender** | | | | | | 0.5 | 0.462 |
| Male | 14631 | 4903 | 34.0 (30.0–39.0) | 2579.3 | 97.3 | | |
| Female | 54190 | 20278 | 34.9 (30.4–38.4) | 10535.6 | 98.6 | | |
| **Residence** | | | | | | 60.2 | |
| Rural | 14294 | 4894 | 35.1 (30.5–39.9) | 2704.5 | 97.0 | | <0.001 |
| Urban | 17635 | 5374 | 31.0 (27.0–35.0) | 2590.6 | 97.0 | | |
| **Age** | | | | | | 788.7 | <0.001 |
| ≤ 10 y | 2799 | 542 | 17.1 (11.9–23.1) | 320.5 | 92.5 | | |
| 11–20 y | 8378 | 2182 | 25.6 (21.7–29.9) | 1050.9 | 93.7 | | |
| 21–50 y | 22174 | 7765 | 36.0 (31.0–40.0) | 3526.7 | 98.1 | | |
| ≥ 50 y | 2832 | 1301 | 44.4 (31.9–57.2) | 1279.5 | 97.7 | | |
| **Educational level** | | | | | | 187.2 | <0.001 |
| Illiterate | 2271 | 760 | 45.3 (34.4–56.5) | 730.1 | 95.8 | | |
| Primary | 5124 | 2040 | 39.4 (34.7–42.2) | 465.4 | 91.2 | | |
| High school | 8935 | 3170 | 34.0 (30.0–39.0) | 1078.1 | 95.1 | | |
| University | 5405 | 2018 | 30.9 (25.8–36.2) | 887.6 | 94.0 | | |
| **Occupation** | | | | | | 135.6 | <0.001 |
| Unemployed | 252 | 40 | 38.6 (23.5–83.4) | 69.2 | 95.7 | | |
| Employee | 2134 | 868 | 36.0 (29.0–44.0) | 448.2 | 92.4 | | |
| Self-employment | 1596 | 527 | 35.9.0 (26.8–45.7) | 262.6 | 93.1 | | |
| Housewife | 8005 | 3171 | 42.0 (35.0–48.0) | 1262.4 | 97.3 | | |
| Student | 2112 | 553 | 29.9 (23.2–37.1) | 235.7 | 91.3 | | |
| Farmer | 708 | 257 | 46.6 (28.9–64.9) | 258.1 | 95.3 | | |
| **Type of population** | | | | | | 614.67 | <0.001 |
| Blood donor | 9015 | 2887 | 33.00 (26.00–40.00) | 1256.8 | 98.20 | | |
| Children | 5582 | 1211 | 16.48 (9.50–24.93) | 980.2 | 98.30 | | |
| General | 65117 | 21124 | 33.60 (30.70–36.50) | 11200.4 | 98.40 | | |
| Pregnant women | 41908 | 15484 | 35.00 (31.00–39.00) | 6356.9 | 98.40 | | |

individuals at 45.3% (34.4–56.5%). Comparatively, those with primary, high, and university education exhibited anti-*T. gondii* IgG seroprevalences of 39.4% (34.7–42.2%), 34% (30–39%), and 30.9% (25.8–36.2%), respectively (Table 3). Furthermore, occupation-based analysis revealed the highest anti-*T. gondii* IgG seroprevalence among farmers at 46.6% (28.9–64.9%). Subsequently, housewives (42%; 35–48%), unemployed individuals (38.6%; 23.5–83.4%), employees (36%; 29–44%), self-employed individuals (35.9%; 26.8–45.7%), and students (29.9%; 23.2–37.1%) showed descending prevalence rates (Table 3). Despite women exhibited a slightly higher pooled seroprevalence of toxoplasmosis at 34.9% (95% CI: 30.4–38.4%) compared to men at 34% (95% CI: 30–39%), this gender difference was not significant (P = 0.462) (Table 3).

## Subgroup analysis of risk factors

The studies were appraised using the JBI quality assessment, and we rated 142 (43.43%) studies as high quality and 146 (44.65%) studies as medium quality; other studies were rated as low quality. The average quality assessment score was 5.87±0.179 which indicated moderate quality. Analyzing the type of population, the highest seroprevalence of IgG anti-*T. gondii* was observed among pregnant women at 35.00 (31.00–39.00). In contrast, children had the lowest prevalence, 16.48 (9.50–24.93). Also, blood donors and general groups had similar prevalence percentages and showed 33.00 (26.00–40.00) and 33.60 (30.70–36.50), respectively (Table 3). The comparison of studies based on the type of study included Cross-sectional, Case-control,

**Table 4. Subgroup analysis of anti-*Toxoplasma gondii* IgG seroprevalence in the general population of Iran according to potential risk factors of exposition (2000 to 2023).**

| Subgroup variable | No. samples | No. seropositive | Prevalence % (95% CI) | Heterogeneity (Q) | $I^2$ (%) | Interaction test ($X^2$ value) | P-value |
|---|---|---|---|---|---|---|---|
| **Contact with cat** | | | | | | | |
| Yes | 9293 | 3495 | 34.1 (29.9–38.4) | 1665.6 | 94.4 | 258.1 | <0.001 |
| No | 19445 | 5497 | 25.0 (21.0–29.0) | 3904.9 | 97.7 | | |
| **Eating undercook meat** | | | | | | | |
| Yes | 8512 | 2992 | 36.4 (31.4–41.5) | 1568.8 | 95.6 | 197.4 | <0.001 |
| No | 12652 | 3667 | 27.3 (22.6–32.3) | 2484.7 | 97.4 | | |
| **Eating raw vegetables** | | | | | | | |
| Yes | 9196 | 3474 | 34.2 (27.0–41.9) | 2074.6 | 98.3 | 33.1 | <0.001 |
| No | 3168 | 877 | 28.7 (22.0–35.8) | 621.4 | 94.5 | | |
| **Washing of vegetables** | | | | | | | |
| Just water | 4275 | 1398 | 32.0 (23.0–42.0) | 942.3 | 97.6 | 3.4 | 0.065 |
| Using detergent | 2836 | 1137 | 32.8 (22.5–43.9) | 849.3 | 97.4 | | |
| **Water source** | | | | | | | |
| Purified | 4362 | 1602 | 32.0 (23.0–41.0) | 858.3 | 97.7 | 65.1 | <0.001 |
| Unpurified | 772 | 298 | 47.0 (33.0–61.0) | 232.2 | 91.8 | | |

and Cohort, which were 250 papers (76.45%), 75 papers (22.94%), and 2 papers (0.61%), respectively.

In the subgroup analysis regarding contact with cats, individuals reporting contact with cats showed a higher anti-*T. gondii* IgG seroprevalence at 34.1% (29.9–38.4%) compared to those without such contact (25%; 21–29%) (P<0.001) (Table 4). Regarding dietary habits, individuals reporting the consumption of undercooked meat had a higher record (36.4%; 31.4–41.5%) compared to those who indicated to have consumed well-cooked meat (27.3%, 22.6–32.3%) (P<0.001) (Table 4). Similarly, those who reported intake of raw vegetables exhibited a higher seroprevalence (34.2%, 27–41.9%) compared to those not consuming (28.7%; 22.0–35.8%) (P<0.001) (Table 4). Regarding the use of water sources, groups consuming unpurified water showed a higher record seroprevalence (47%, 33–61%) compared to those reporting drinking purified water (32%, 23–41%) (P = 0.065) (Table 4). Statistically significant differences were observed in all subgroups of risk factors (P<0.001), except for seroprevalence level in the subgroup about the method used to wash vegetables (P = 0.065) (Table 4).

## Discussion

In this systematic review and meta-analysis, an ambitious effort has been done to assess the anti-*T. gondii* IgG seroprevalence in the general population between 2000 to 2023; overall data indicate that approximately one-third (33%) of the individuals tested have been exposed to *T. gondii*. Our study improves upon the previous meta-analysis study [14], encompassing a larger sample size (122,882 cases) covering all Iranian provinces over twenty-four more recent years. Seroprevalence in healthy individuals worldwide typically ranges between 0.5% and 87.7%, with African countries demonstrating the highest prevalence at 61.4%, while North American and Far East Asian countries have the lowest rates [1]. Pooled seroprevalence data reported here remarkably aligns with the global and European average [20] and is comparable to neighboring countries like Saudi Arabia [21]. A previous report from 2014 estimated a 39% seroprevalence [14], small differences may be due to potential epidemiological changes (eg., public awareness) or a broader number of studies considered. We identified regional disparities within Iran, notably higher anti-*T. gondii* IgG seroprevalence records in northern provinces, and conversely, central and southern regions with lower rainfall, humidity, and higher

temperatures. However, limited studies in some provinces may not fully represent the prevalence of toxoplasmosis in such areas (Table 1).

Geographical differences seem to be correlated to higher humidity level and precipitation records. Conversely, higher temperatures were linked to lower anti-*T. gondii* IgG seroprevalences 23.0% (20.0–26.0) (Table 2). These findings emphasize the impact of environmental conditions on *T. gondii* transmission and distribution. Oocysts survive longer in mild, humid climates, increasing the chances of transmission to new hosts. Even slight temperature fluctuations significantly affect oocyst transmission [22]. These findings align with previous meta-analyses, showing elevated anti-*T. gondii* IgG seroprevalence in northern Iranian provinces [14]. Additionally, observations in Sweden (Yan et al., 2016) [22] further support our results, demonstrating a positive correlation between average annual temperatures and *T. gondii* incidence in pregnant women. In addition, rainfall fosters oocyst survival by creating moist environments but also aids in transferring oocysts from land to water, potentially causing waterborne *T. gondii* infections [23,24].

Our subgroup analysis of anti-*T. gondii* IgG seroprevalence trends in Iran since 2000 revealed a consistent pattern until a significant decline after 2021 (Table 2). This decrease aligned with the global emergence of the COVID-19 pandemic, coinciding with potential adherence to health protocols during this period [25]. Factors such as improved public health education leading to behavioral changes, increased handwashing, fruit and vegetable cleaning, and healthier cooking practices likely contributed to reducing the spread of infectious diseases, including toxoplasmosis, from 2021 onwards [26–29].

We found a higher anti-*T. gondii* IgG seroprevalence among rural dwellers compared to urban residents in Iran (Table 3) is potentially linked to rural activities involving soil, livestock, poultry, and consumption of contaminated vegetables and untreated water [30]. Our study demonstrated a positive correlation between age and anti-*T. gondii* IgG seroprevalence in the GPI (Table 3), attributed to cumulative seropositivity due to prolonged exposure to risk factors [31], corroborating previous findings from previous studies [14,31,32]. Lower education levels correlate with higher *T. gondii* prevalence, potentially associated with occupations involving soil exposure, lack of disease knowledge, and poorer health conditions [31,33]. Our study corroborates this finding, showing decreasing IgG seroprevalence with increased education levels in the GPI. Notably, illiterate individuals exhibited a higher prevalence, whereas those with university education showed the lowest rates (Table 3). Farmers and housewives displayed the highest anti-*T. gondii* IgG seroprevalence among various occupations in Iran, consistent with previous findings [14,34]. Farmers, due to soil exposure, and housewives, especially during food preparation, appear more susceptible to *T. gondii* infection [34]. Educating housewives on preventive measures during food preparation is crucial to curbing *T. gondii* transmission [14,35].

Consistent with previous studies [14,36], our findings indicated that the population's contact with domestic cats significantly influences *T. gondii* infection distribution. With a rise in pet ownership in Iran, this trend could contribute to increased *T. gondii* infection rates in the future. As well as household cats can represent a direct risk for owners [37]. Felines, through environmental contamination, significantly contribute to *T. gondii* transmission. Accidental ingestion of oocysts via contaminated products like water and raw fresh produce plays a crucial role in human infection [14,38] as observed herein. Consumption of raw or undercooked meat, particularly pork and sheep, poses a risk for *T. gondii* infection, with organic livestock showing higher contamination rates than industrially bred animals due to open-space access [38–40]. Consistent with prior studies, our results underscored higher prevalence in individuals consuming raw or undercooked meat products (Table 4), emphasizing the need to avoid these in high-risk groups, especially pregnant women [40[. Also, this study observed the

highest seroprevalence among pregnant women. Considering that the risk of contracting this parasite during pregnancy is clinically important, we suggest that preventive measures to prevent pregnant women from becoming infected with *T. gondii* should be strictly applied. However, it should be noted that infection before pregnancy causes resistance to infection and transmission to the fetus. As a result, IgM identification is important in pregnant women [41]. Additionally, there are no significant gender-based differences in anti-*T. gondii* IgG seroprevalence was observed, similar to previous studies [14].

A fact that deserves attention is the potential impact of culinary differences between regions. Dietary/culinary habits play a role in anti-*T. gondii* IgG seroprevalence, and may partially explain regional differences within Iran; particularly in the north of Iran, where higher-income areas witness more consumption of semi-cooked and grilled meats. Previous Iranian studies identified mutton consumption, especially in kebab form, as a risk factor for higher anti-*T. gondii* IgG seroprevalence [42]. Additionally, higher anti-*T. gondii* IgG seroprevalence in some regions like Lorestan province, with unsuitable temperatures for oocyst survival, may be correlated with increased consumption of undercooked meat, particularly barbecued kebabs. Moreover, migration from rural to urban areas due to economic changes has altered *T. gondii* prevalence, as seen in Qom province, which despite its hot and dry climate, exhibits a relatively high prevalence due to recent population growth [43].

Our study improves upon the previous meta-analysis study [14], encompassing a larger sample size (122,882 cases) covering all Iranian provinces over nine more recent years. We also performed various subgroups, including geo-climate variables and water sources.

## Limitations

limitations included the uneven distribution of studies across provinces, varying diagnostic methods' sensitivity and specificity, impacting prevalence accuracy, and substantial heterogeneity between studies.

## Conclusions

In conclusion, our meta-analysis, encompassing all provinces in Iran, revealed that approximately one-third of the GPI were exposed to *T. gondii*, while two-thirds remain susceptible. Given the opportunistic character of *T. gondii*, monitoring at-risk population segments, especially pregnant women and immunosuppressed individuals using harmonized methodologies is crucial. Nevertheless, surveillance systems should be accompanied by implementing health education programs, setting higher standards in the food industries (through revised regulation), and improving animal health schemes are essential items in the search for a One Health approach against *Toxoplasma* infections.

## Supporting information

**S1 File.** S1 Fig. Funnel plot of standard error by logit event rate to illustrate assessment of publication bias for studies reporting anti-*Toxoplasma gondii* IgG seroprevalence in the general population of Iran. S2 Fig. Random-effect meta-analysis of pooled estimation of Anti-*Toxoplasma gondii* IgG seroprevalence in the general population in Iran. S1 Table. Main characteristics of the included studies.
(DOC)

## Acknowledgments

The authors thank the Infectious Diseases Research Center, Gonabad University of Medical Science, for their support, and also appreciate and thank the anonymous reviewers for their comments.

## Author Contributions

**Conceptualization:** Ali Rostami, Seyed Abdollah Hosseini, Hossein Pazoki.

**Data curation:** Faezeh Hamidi, Ali Rostami, Jafar Hajavi, Hossein Pazoki.

**Formal analysis:** Seyed Abdollah Hosseini, Hossein Pazoki.

**Investigation:** Faezeh Hamidi, Ali Rostami, Hossein Pazoki.

**Methodology:** Faezeh Hamidi, Seyed Abdollah Hosseini, Hossein Pazoki.

**Supervision:** Hossein Pazoki.

**Writing – original draft:** Faezeh Hamidi, Ali Rostami, Seyed Abdollah Hosseini, Reza Ahmadi, Hossein Pazoki.

**Writing – review & editing:** Ali Rostami, Rafael Calero-Bernal.

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
