## [Decision Letter · Decision Letter 0]

27 May 2024

PONE-D-24-16334Anti-Toxoplasma gondii IgG seroprevalence in the general population in Iran: A systematic review and meta-analysis, 2000–2023PLOS ONE

Dear Dr. pazoki,

Thank you for submitting your manuscript to PLOS ONE. After careful consideration, we feel that it has merit but does not fully meet PLOS ONE’s publication criteria as it currently stands. Therefore, we invite you to submit a revised version of the manuscript that addresses the points raised during the review process.

We look forward to receiving your revised manuscript.

Kind regards,

Masoud Foroutan, Ph.D; Assistant Professor

Academic Editor

PLOS ONE

Journal Requirements:

5. We note that Figure 2 in your submission contain map/satellite images which may be copyrighted. All PLOS content is published under the Creative Commons Attribution License (CC BY 4.0), which means that the manuscript, images, and Supporting Information files will be freely available online, and any third party is permitted to access, download, copy, distribute, and use these materials in any way, even commercially, with proper attribution. For these reasons, we cannot publish previously copyrighted maps or satellite images created using proprietary data, such as Google software (Google Maps, Street View, and Earth). For more information, see our copyright guidelines: http://journals.plos.org/plosone/s/licenses-and-copyright.

6. Please include your tables as part of your main manuscript and remove the individual files. Please note that supplementary tables (should remain/ be uploaded) as separate "supporting information" files

Reviewers' comments:

Reviewer's Responses to Questions

**Comments to the Author**

1. Is the manuscript technically sound, and do the data support the conclusions?

Reviewer #1: Yes

Reviewer #2: Yes

2. Has the statistical analysis been performed appropriately and rigorously? 

Reviewer #1: Yes

Reviewer #2: Yes

3. Have the authors made all data underlying the findings in their manuscript fully available?

Reviewer #1: Yes

Reviewer #2: Yes

4. Is the manuscript presented in an intelligible fashion and written in standard English?

Reviewer #1: Yes

Reviewer #2: Yes

5. Review Comments to the Author

Reviewer #1: Thank you for selecting me to review this valuable article. First of all, I would like to thank the authors for the excellent design, strong and clear writing, and the new analyses presented in this article. I have only a few minor comments:

1- The scientific name should be italicized both in the main text and in the references.

2- In the Introduction section, the role of consuming animal products such as poultry, dairy, and eggs should be better explained.

3- The Limitations section should have a separate subheading.

Reviewer #2: The manuscript " Anti-Toxoplasma gondii IgG seroprevalence in the general population in Iran: A systematic review and meta-analysis, 2000–2023" presented the anti-T. gondii IgG seroprevalence within the general population in Iran, a large country in the Middle East. These article kinds are significant for increasing the sensitivity of the medical community and health policymakers especially in developing countries.

The article structure is acceptable, but the authors should answer the questions and make changes based on the comments.

Abstract

- The abstract should not be presented in a structured format

- " Chronic toxoplasmosis is associated with several neuropsychiatric and other harmful effects in infected people, emphasizing the need to assess its burden across various world regions.” → It is incorrect to mention the facts in the proof and hypothesis stage as absolutely proven.

Keywords

- Keywords must be in MeSH (Medical Subject Headings) term.

- Keywords must start with a capital letter

- Keywords must be given in alphabetical order.

Introduction

- Using appropriate and detailed references in all parts of the manuscript. In a review article, referring to other review articles must be done with reason and logic, otherwise, it is unacceptable.

- References 2 to 5 should be removed, because they are unrelated to text and will lead to self-citation, which is against publication ethics.

- “Recent studies have proposed links between chronic toxoplasmosis infection and psychiatric disorders like schizophrenia [10], depression [9], bipolar disorder [10], and epilepsy [11], underscoring the importance of reliable diagnosis of chronic toxoplasmosis within the general population.“ → Summarize and use original articles for reference.

- “The exposition of T. gondii in the Iranian population has been mostly approached by the detection of anti-Toxoplasma IgG by serological methods like enzyme-linked immunosorbent assay (ELISA), and indirect immunofluorescence antibody test (IFA)” → Applied methods in Iran have not been fully addressed

- “While several meta- analysis studies focused on T. gondii occurrence in immunocompromised individuals such as cancer patients [6, 13-15]” → There is no need to mention other meta-analyses and different study groups. Be omitted.

- "IGP" In the last sentence of the introduction → Be corrected

Methods

- “The search terms used encompassed "Toxoplasma gondii", "T. gondii", "toxoplasmosis", "Iran", "Islamic Republic of Iran", "seroprevalence", and "general population".” → It is necessary to mention the strategic syntax search

- “general population” is not MeSH term

- A number of keywords are missed: ‘Toxoplasma”, “seropositivity” and “prevalence”

Results

- Considering in the analysis table, the results of the cities of Mazandaran, Khuzestan, Qom, Kurdistan, Gilan, Kermanshah, Qazvin, Lorestan, Yazd, Fars, Hormozgan, Semnan, Kerman, and Sistan and Baluchistan have more different results compared to Kalantari et al. that recently published study and you just mention that in the introduction. Are the results of this study not included in the above analysis? What argument would you make on this?

Discussion

- Reference 22 is not relevant and needs to be removed

- Reference 33 is not relevant and needs to be removed

- “Also, in this study, the highest seroprevalence was observed among pregnant women. Considering that the risk of contracting this parasite during pregnancy is clinically important [45], we suggest that preventive measures to prevent pregnant women from becoming infected with T. gondii should be strictly applied. However, it should be noted that infection before pregnancy causes resistance to infection and transmission to the fetus. As a result, IgM identification is important in pregnant women [45].” → Use original articles for reference, also, considering the main topic of the article, it is not relevant to deal with pregnant women.

First, the "Disclosures" section should be mentioned, and then "Compliance with Ethics Guidelines" should be mentioned.

References

- The second reference must be correctly given. This book is not published in 2020.

6. PLOS authors have the option to publish the peer review history of their article (what does this mean?). If published, this will include your full peer review and any attached files.

Reviewer #1: No

Reviewer #2: No

---

## [Author Response · Author response to Decision Letter 0]

4 Jul 2024

Dear editor in chief;

We thank you very much for your efforts in reviewing our manuscript and valuable comments and suggestions. I'm sharing our responses to your comments. The changes made in the "Revised Manuscript with Track Changes" were highlighted for the editor with blue color, Reviver 1 with green color, and Reviver 2 with yellow color.

Corresponding author: Dr. Hossein Pazoki,

Email: hosseinpazoki11@gmail.com;

Tel: +91289122072

Comments and Responses: 

1. Please ensure that your manuscript meets PLOS ONE's style requirements, including those for file naming. The PLOS ONE style templates can be found at https://journals.plos.org/plosone/s/file?id=wjVg/PLOSOne_formatting_sample_main_body.pdf and https://journals.plos.org/plosone/s/file?id=ba62/PLOSOne_formatting_sample_title_authors_affiliations.pdf.

Response: According to your comment, the manuscript was adapted to PLOS ONE style requirements.

Response: Financial aid information is matched in the "Funding Information" and "Financial Disclosure" sections.

3. We note that your Data Availability Statement is currently as follows: All relevant data are within the manuscript and its Supporting Information files. Please confirm at this time whether or not your submission contains all raw data required to replicate the results of your study. Authors must share the “minimal data set” for their submission. PLOS defines the minimal data set to consist of the data required to replicate all study findings reported in the article, as well as related metadata and methods (https://journals.plos.org/plosone/s/data-availability#loc-minimal-data-set-definition).

Response: We certify that our submission contains all the raw data required to replicate the results of this study.

Response: All of our data is freely available.

5. We note that Figure 2 in your submission contain map/satellite images which may be copyrighted. All PLOS content is published under the Creative Commons Attribution License (CC BY 4.0), which means that the manuscript, images, and Supporting Information files will be freely available online, and any third party is permitted to access, download, copy, distribute, and use these materials in any way, even commercially, with proper attribution. For these reasons, we cannot publish previously copyrighted maps or satellite images created using proprietary data, such as Google software (Google Maps, Street View, and Earth). For more information, see our copyright guidelines: http://journals.plos.org/plosone/s/licenses-and-copyright.

Response: According to your comment, the authors confirm that this map (Fig 2) is without any copyright. This figure was drawn by the authors of this article using ArcGIS version 10.5 software and then edited by Adobe Photoshop 2024 software and its quality has been improved.

6. Please include your tables as part of your main manuscript and remove the individual files. Please note that supplementary tables (should remain/ be uploaded) as separate "supporting information" files

Response: Our tables were placed in the main manuscript and removed from separate files.

Response to Reviewer #1:

1- The scientific name should be italicized both in the main text and in the references.

Response: According to your valuable comments, the scientific name was italicized both in the main text and in the references.

2. In the Introduction section, the role of consuming animal products such as poultry, dairy, and eggs should be better explained.

Response: Thank you for your precise comment. Initially, when writing the article, the parasite cycle was fully explained in the introduction section, but due to the many results of this article, the length of the discussion, and the limitation of the number of words, we had to remove many sections of the introduction.

In the introduction, we did not explain anything about the life cycle of Toxoplasma and the role of consumption of animal products, and if we explain the role of consumption of animal products such as chicken, dairy, and eggs, then the rest of the cycle should also be explained. So, if you allow this section not to be added.

3. The Limitations section should have a separate subheading.

Response: According to your valuable comments, the Limitations section was separated and highlighted. 

Response to Reviewer #2

 1. The abstract should not be presented in a structured format

Response: Yes. The Abstract was corrected and highlighted. 

2. " Chronic toxoplasmosis is associated with several neuropsychiatric and other harmful effects in infected people, emphasizing the need to assess its burden across various world regions.” → It is incorrect to mention the facts in the proof and hypothesis stage as absolutely proven.

Response: Yes, your comment is absolutely right, the sentence was modified and highlighted. 

#Keywords 

- Keywords must be in MeSH (Medical Subject Headings) term.

- Keywords must start with a capital letter

- Keywords must be given in alphabetical order.

Response: According to your valuable comments, keywords were corrected and highlighted. 

# Introduction

4. Using appropriate and detailed references in all parts of the manuscript. In a review article, referring to other review articles must be done with reason and logic, otherwise, it is unacceptable.

Response: They were corrected and used original articles and highlighted. 

5. References 2 to 5 should be removed, because they are unrelated to text and will lead to self-citation, which is against publication ethics.

Response: The references were removed. 

6. “Recent studies have proposed links between chronic toxoplasmosis infection and psychiatric disorders like schizophrenia [10], depression [9], bipolar disorder [10], and epilepsy [11], underscoring the importance of reliable diagnosis of chronic toxoplasmosis within the general population. “→ Summarize and use original articles for reference.

Response: It was summarized and used original articles for reference (References 7,8,9)

7. “The exposition of T. gondii in the Iranian population has been mostly approached by the detection of anti-Toxoplasma IgG by serological methods like enzyme-linked immunosorbent assay (ELISA), and indirect immunofluorescence antibody test (IFA)” → Applied methods in Iran have not been fully addressed

Response: Suitable references were added to the end of the sentence (references 10 and 11) 

8. “individuals such as cancer patients [6, 13-15]” → There is no need to mention other meta-analyses and different While several meta-analysis studies focused on T. gondii occurrence in immunocompromised study groups. Be omitted.

Response: Our aim from this sentence was to emphasize that most research is on immunocompromised individuals and Toxoplasma research is deficient in the general population. 

9. "IGP" In the last sentence of the introduction → Be corrected

Response:, It was corrected and highlighted. 

#Methods

10. “The search terms used encompassed "Toxoplasma gondii", "T. gondii", "toxoplasmosis", "Iran", "Islamic Republic of Iran", "seroprevalence", and "general population".” → It is necessary to mention the strategic syntax search

Response: It is done and highlighted. 

11. “general population” is not MeSH term. 

Response: It was omitted. 

12. A number of keywords are missed: ‘Toxoplasma”, “seropositivity” and “prevalence”

Response: We have searched articles by these terms but it was forgotten adding our manuscript. So, they were added to the manuscript. 

#Results

13. Considering in the analysis table, the results of the cities of Mazandaran, Khuzestan, Qom, Kurdistan, Gilan, Kermanshah, Qazvin, Lorestan, Yazd, Fars, Hormozgan, Semnan, Kerman, and Sistan and Baluchistan have more different results compared to Kalantari et al. that recently published study and you just mention that in the introduction. Are the results of this study not included in the above analysis? What argument would you make on this?

Response: Thank you for your precise comment but there are differences between our study and Kalantari et al.

1. our sample size is 122882 and Kalantari et al is 35047 individuals. 

2. our study was conducted between 2000 and 2023 but Kalantari et al ‘study was conducted between 2015 and 2020. So, our study period is longer than their study. Therefore, it is normal that the long interval and larger sample size affect the percentage of Toxoplasma prevalence in different cities. 

 3. Meta-analysis was not performed in the study of Kalantari et al.

It should be noted that their studies were included in our analysis. 

#Discussion

14. Reference 22 is not relevant and needs to be removed. 

Response: It was removed. 

15. Reference 33 is not relevant and needs to be removed. 

Response: It was removed. 

16. “Also, in this study, the highest seroprevalence was observed among pregnant women. Considering that the risk of contracting this parasite during pregnancy is clinically important [45], we suggest that preventive measures to prevent pregnant women from becoming infected with T. gondii should be strictly applied. However, it should be noted that infection before pregnancy causes resistance to infection and transmission to the fetus. As a result, IgM identification is important in pregnant women [45].” → Use original articles for reference, also, considering the main topic of the article, it is not relevant to deal with pregnant women.

Response: The original reference was added and highlighted (reference 41).

17. First, the "Disclosures" section should be mentioned, and then "Compliance with Ethics Guidelines" should be mentioned. 

Response: These subheadings were added to the manuscript and highlighted. 

#References

18. The second reference must be correctly given. This book is not published in 2020.

Response: Thank you for your precise comment. This book was published in 2014. You asked us to omit this reference from the introduction section. So, it was omitted.

---

## [Decision Letter · Decision Letter 1]

16 Jul 2024

Anti-Toxoplasma gondii IgG seroprevalence in the general population in Iran: A systematic review and meta-analysis, 2000–2023

PONE-D-24-16334R1

Dear Dr. pazoki,

We’re pleased to inform you that your manuscript has been judged scientifically suitable for publication and will be formally accepted for publication once it meets all outstanding technical requirements.

Kind regards,

Masoud Foroutan, Ph.D; Assistant Professor

Academic Editor

PLOS ONE

Additional Editor Comments (optional):

Reviewers' comments:

Reviewer's Responses to Questions

**Comments to the Author**

1. If the authors have adequately addressed your comments raised in a previous round of review and you feel that this manuscript is now acceptable for publication, you may indicate that here to bypass the “Comments to the Author” section, enter your conflict of interest statement in the “Confidential to Editor” section, and submit your "Accept" recommendation.

Reviewer #1: All comments have been addressed

Reviewer #2: All comments have been addressed

2. Is the manuscript technically sound, and do the data support the conclusions?

Reviewer #1: Yes

Reviewer #2: Yes

3. Has the statistical analysis been performed appropriately and rigorously? 

Reviewer #1: Yes

Reviewer #2: Yes

4. Have the authors made all data underlying the findings in their manuscript fully available?

Reviewer #1: (No Response)

Reviewer #2: Yes

5. Is the manuscript presented in an intelligible fashion and written in standard English?

Reviewer #1: (No Response)

Reviewer #2: Yes

6. Review Comments to the Author

Reviewer #1: The authors have answered all of my questions and the paper has been greatly improved. Therefore, it can be accepted for publication.

Reviewer #2: (No Response)

7. PLOS authors have the option to publish the peer review history of their article (what does this mean?). If published, this will include your full peer review and any attached files.

Reviewer #1: No

Reviewer #2: No

---

## [Editor Report · Acceptance letter]

21 Aug 2024

PONE-D-24-16334R1 

PLOS ONE

Dear Dr. Pazoki, 

I'm pleased to inform you that your manuscript has been deemed suitable for publication in PLOS ONE. Congratulations! Your manuscript is now being handed over to our production team.

Kind regards, 

on behalf of

Dr. Masoud Foroutan 

Academic Editor

PLOS ONE